# The Reentry Health Care Hub: Creating a California-Based Referral System to Link Chronically Ill People Leaving Prison to Primary Care

**DOI:** 10.3390/ijerph20105806

**Published:** 2023-05-12

**Authors:** Bethany Divakaran, Natania Bloch, Mahima Sinha, Anna Steiner, Shira Shavit

**Affiliations:** 1San Francisco Public Health Foundation Project, Transitions Clinic Network, San Francisco, CA 94102, USA; 2School of Public Health, University of California-Berkeley, Berkeley, CA 94704, USA; 3School of Community Family Medicine, University of California-San Francisco, San Francisco, CA 94103, USA

**Keywords:** reentry health care, prison, primary care, Transitions Clinic Network, California, CalAIM

## Abstract

People released from prison experience high health needs and face barriers to health care in the community. During the COVID-19 pandemic, people released early from California state prisons to under-resourced communities. Historically, there has been minimal care coordination between prisons and community primary care. The Transitions Clinic Network (TCN), a community-based non-profit organization, supports a network of California primary care clinics in adopting an evidence-based model of care for returning community members. In 2020, TCN linked the California Department of Corrections and Rehabilitation (CDCR) and 21 TCN-affiliated clinics to create the Reentry Health Care Hub, supporting patient linkages to care post-release. From April 2020–August 2022, the Hub received 8420 referrals from CDCR to facilitate linkages to clinics offering medical, behavioral health, and substance use disorder services, as well as community health workers with histories of incarceration. This program description identifies care continuity components critical for reentry, including data sharing between carceral and community health systems, time and patient access for pre-release care planning, and investments in primary care resources. This collaboration is a model for other states, especially after the Medicaid Reentry Act and amid initiatives to improve care continuity for returning community members, like California‘s Medicaid waiver (CalAIM).

## 1. Introduction

People who are incarcerated have higher medical, mental health, and substance use disorder (SUD) needs than the general population [1,2,3,4,5]. Tens of thousands of people release from jail and prison in California annually, the majority of whom will require ongoing primary care in the community [6,7,8]. However, the health conditions of returning community members tend to worsen upon release due to challenges with care transitions [9]. The reentry population experiences more emergency room visits and hospitalizations than the general population from issues that could be mitigated by timely access to primary care [10,11]. This population has a 12 times greater risk of death than the general population in the two weeks following release, including an increased risk of mortality from cardiovascular disease and cancer and a 129 times higher risk of death by overdose [12]. Coming home from highly structured carceral health systems, patients can be ill-equipped to manage their medical conditions and navigate complex and fragmented community health systems [11,13]. Despite having collectively greater needs, primary care is underutilized among returning community members [11,14].

Returning community members often emerge from incarceration to communities that lack sufficient health and reentry supports to meet their complex medical and social needs [2,11,15]. Furthermore, patients’ experiences of stigma and discrimination within carceral and community health care systems impact their engagement in health care services [16,17,18]. One study found that patients who identify as formerly incarcerated were half as likely to obtain a new patient primary care appointment as people without histories of incarceration [17]. Barriers to accessing health care impede employability, prevent financial gain and housing stability, and increase the risk of recidivism [2,4,19].

There is also a gap in health care insurance coverage as patients transition from carceral health care to community-based coverage, which contributes to the challenges of care continuity. Due to the federally implemented Medicaid Inmate Exclusion Policy, people who are incarcerated are excluded from Medicaid coverage [19]. Depending on the state in which one lives, insurance coverage through Medicaid is suspended or terminated when one is incarcerated, and patients cannot reactivate Medicaid until they have returned to the community [20,21]. Patients leave incarceration with limited or no supply of medications and medical supplies, rendering linkage to community care time sensitive. However, a gap in insurance upon reentry may be a barrier to timely connection to a primary care clinic for medication refills and ongoing care [20]. These gaps in care are especially problematic for patients with high-risk medical conditions who require prompt interventions and specialty care [22].

Given prison and community health systems are disjointed, there is a critical need to improve the continuity of care between systems to mitigate harms during this high-risk period following release from incarceration. While carceral institutions are federally mandated to provide basic medical care to incarcerated people, there is no legal mandate for providing continuity of care at release. Most jail and prison systems lack the infrastructure for coordinating the transition of medical information and services to the community for individuals with chronic conditions. Community health systems are not incentivized or equipped to provide pre-release services and are typically limited to serving specific geographic jurisdictions rather than coordinating services statewide [14]. Best practices in release planning include partnerships between carceral and community health systems to ensure continuity of health care and to avoid the duplication of services [23]. These types of partnerships are sporadic, and patients are often left to navigate service gaps by themselves.

The size of California and the complexity of its systems necessitates systematic ways to coordinate care. California has the second largest prison system in the United States, with 33 statewide prison facilities [24,25] resulting in many people being incarcerated far from their homes. Individuals are typically mandated to return to their last county of residence, and medical and social resources vary widely across California’s 58 counties. In California, there are a growing number of federally qualified health centers (FQHCs) or FQHC look-a-like clinics. As of January 2023, over 200 FQHCs operate at over 2700 delivery sites across the state [26]. Most of these primary health systems operate independent of each other and are not coordinated in providing services or sharing patient information, the result being that patients may receive medical services from more than one disjointed system. Community clinics do not routinely receive information from the prison system for patients returning to the community. Further, as 88% percent of people served by Medicaid are enrolled in managed care, clinics often must navigate relationships with many different payers [27]. FQHCs vary in size, the services offered, and the ability to meet the myriad of complex needs returning community members may have [28].

In the 2019–2020 fiscal year, the California Department of Corrections and Rehabilitation (CDCR) received state funding to start the Integrated Substance Use Disorder Treatment (ISUDT) Program to address the disproportionately high rate of prison opioid overdose [29]. The ISUDT program goals include the treatment of people with opioid use disorder (OUD) with medications for opioid use disorder (MOUD) and care coordination to support continuity of care into the community upon release. Each state prison hired resource nurses to coordinate treatment inside prison and post release, establishing the first dedicated health care position for care coordination.

In 2020, COVID-19 spread through US state and federal prisons at a rate four times higher than the national average, with twice as many mortalities [30]. California prisons and jails also reported a disproportionately high infection rate for those incarcerated during the pandemic [31]. At the urging of health professionals and advocates, people were released from California prisons early if already pending release or meeting criteria for medical release, in order to reduce mortality from COVID-19 amongst vulnerable populations [32,33]. These individuals were rapidly released often to resource-poor health systems impacted even further by pandemic stressors. At that time, the need for system improvements in care coordination from prison to community health systems had never been clearer.

The Transitions Clinic Network (TCN) is a non-profit organization that builds the capacity of health systems to improve the health and well-being of people impacted by incarceration. Primary care clinics in communities disproportionately impacted by incarceration integrate TCN’s evidence-based model of care into their practice by hiring community health workers (CHWs) with lived experience of incarceration as part of their care teams. CHWs work closely with chronically ill community members returning from incarceration to provide education, health care navigation, and advocacy to address health and social needs, including housing and employment. Clinics also transform their practice by providing clinical services targeted towards returning community members’ health needs, such as integrated behavioral health services, trauma-informed care, and hepatitis C treatment. This model of care goes beyond providing individual-level patient-centered clinical services to also transform health systems, establishing cultures of inclusion, engagement, and collaboration such as through changing hiring practices to employ people with criminal records as CHWs. The TCN model was developed in 2006 in San Francisco through partnership with community stakeholders impacted by incarceration. It has since been implemented in 48 health systems in 14 states and Puerto Rico. In California, there are 21 TCN-affiliated clinics across 14 counties serving patients returning from prison or jail. Past research has demonstrated that patients served by TCN clinics experience half as many preventable emergency room visits and hospitalizations, as well as shorter hospital stays and fewer days reincarcerated [16,34]. A study of one TCN program demonstrated significant savings in criminal-legal system costs without spending more in Medicaid dollars (estimated as a USD 2.55 return of investment to the state for every USD 1 invested in the TCN program) [35].

Since 2008, TCN has provided in-reach care coordination services as an independent community organization to individuals incarcerated at San Quentin (SQ) State Prison through a weekly care coordination clinic for chronically ill patients anticipating release. TCN staff linked patients at SQ to care in their network of statewide clinics or to other primary care clinics in the community. At the onset of the pandemic in the spring of 2020, this model of care connection was leveraged and expanded from individuals at one prison to all people incarcerated across the California prison system to create the TCN Reentry Health Care Hub (the “Hub”).

Historically, discharge planning efforts from carceral entities have focused on populations with specific health needs, such as hospice, behavioral health, or HIV. For instance, the Broward County Jails Hospice Program in Florida is a partnership between a local hospice agency and jail system to coordinate releases [36]. The Jail Inreach Project, developed in 2006 in Harris County, Texas, is a collaboration between the county jail system and county entities to establish post-release plans for persons with behavioral health disorders who are experiencing homelessness [37]. A Los Angeles-based intervention called LINK (Linking Inmates to Care in Los Angeles) aimed to enhance engagement in HIV care for individuals leaving county jails through pre- and post-release case management with peer navigators [38]. In Rhode Island, Project Bridge is another model of care coordination and linkage to care for people living with HIV, focusing on the state prison system to coordinate transitions to community-based primary care [39]. The scope of these studies is limited, focused on local interventions, or targeted towards specific populations. However, there is no published literature on larger scale referral systems from prisons for those with broader primary health care needs. Here, we describe the Hub as a first-of-its-kind statewide program focused on health linkage and referral systems for all people incarcerated with any chronic health condition from prisons to community primary care entities across the state.

## 2. Methods

### 2.1. Developing the Hub Referral Program and Workflow

The Hub serves as an intermediary between California state prisons and community-based primary care clinics to improve release planning and medical information sharing. Three key factors facilitated the creation of the program: a new infrastructure of care coordination in prison, TCN’s statewide patient-centered clinic network located in communities disproportionately impacted by incarceration, and the emerging COVID-19 pandemic.

In 2020, at the onset of the pandemic, TCN advocated, alongside community partners, to facilitate linkages for people with chronic medical conditions returning from incarceration to primary care clinics statewide. TCN successfully petitioned to prison leadership to expand care coordination from the MOUD population to all people with primary care needs. TCN also collaborated with program directors and CHWs at TCN-affiliated clinics and other community reentry organizations to adapt practices to meet the needs of returning community members during the COVID-19 pandemic. The TCN team focused heavily on the input of TCN CHWs and people with lived experiences of incarceration to address the unique health concerns of people coming home from prison during a pandemic. In counties where TCN had multiple sites or pre-existing relationships (such as Los Angeles), TCN also received input for program design from county health care and public health departments. TCN-affiliated clinics committed to receiving referrals from the Hub for patients recently released from prison, with the CHW being the first point of contact.

TCN then developed a referral workflow with input from ISUDT resource nurses and TCN affiliated clinics in the community. From these conversations, TCN developed a simple HIPAA-compliant online survey (Qualtrics XM, Montreal, QC, Canada), which is an important tool for the exchange of medical information and records between prisons and community-based primary care clinics in the absence of any other systematized way of sharing vital health information. To respond to the needs of this community identified by CHWs, the Hub also established a reentry health care hotline staffed by CHWs with lived experience of incarceration to serve as a statewide safety-net resource for people before and after release. The hotline is toll-free and staffed Monday–Friday 9 a.m.–5 p.m. to offer health care navigation support to callers who are incarcerated or recently released.

In addition to this referral workflow, the Hub team compiled a robust catalogue of community-based primary care resources across California used to provide patients with up-to-date resources. This statewide catalogue includes primary care clinics if they accept new patients with Medi-Cal and offer ongoing primary care services at a community address that can be provided to a patient. The catalogue notes if clinics provide additional services, including behavioral health, MOUD, HIV services, and gender-affirming care. Some of these clinic locations are affiliated with TCN’s network, meaning they offer care that integrates medical, behavioral health, and substance use treatment services; collaborate broadly with carceral and community partners; and have CHWs with lived experience of incarceration enhancing health care engagement and supporting reentry-related social determinants of health. Table 1 details the unique combination of services included in the TCN model of care.

### 2.2. Description of the Hub Intervention

Resource nurses saw patients who needed primary care follow up for chronic health conditions 90–120 days (or as soon as possible) prior to their prison release to initiate their release plan. After the patient signed a release of information, nurses submitted referral information to the TCN Hub through the Qualtrics XM survey platform, which includes the attachment of medical records. TCN Hub referral coordinators sorted and prioritized all the referrals received by the next release date. They assessed the patient’s location of release in the community and their medical needs to determine an appropriate community clinic for patient referral.

Hub referral coordinators drafted a resource packet for each patient and sent it to the resource nurse as soon as possible prior to the release date. Each patient received a referral letter with tailored information about insurance enrollment and primary care services as well as the flyer for TCN’s reentry health care hotline. All patients were provided information for a primary care clinic in the community they were returning to that could meet their specific health need. Patients referred to TCN-affiliated clinics could sign a release of information (ROI) to share medical records with their assigned clinic and were also given a direct contact number to a TCN CHW to support their reentry. Patients referred outside the TCN network (due to geography or patient preference) were not assigned a TCN CHW nor did they share medical information with any clinic. Resource nurses met with each patient at least once more prior to release to provide referral information.

Resource nurses scheduled post-release clinic appointments for patients, with Hub staff assisting with coordination to community partners as needed. For patients with high-risk, time-sensitive medical needs, Hub staff also assisted with facilitating a pre-release telephone connection between the patient and the TCN CHW to which they were referred. Following a patient’s release, TCN CHWs reached out to them through available means, i.e., contact information, transitional housing, or required parole meetings) in order to engage them in care. See Figure 1 for a visual representation of this workflow.

### 2.3. Data Collection

The Hub collected data on the following variables from the Qualtrics survey responses, as well as patient medical records and correspondence with nurses, and compiled them into an Excel database: patient gender (categorized as male, female, or transgender); patient age (calculated from date of birth); medical risk [40] (defined by CDCR as High, Medium, or Low depending on the severity and complexity of chronic medical conditions, or Unknown if not provided); mental health risk [40] (defined by CDCR as Acute, Severe/Persistent, Mild–Moderate, None, or Unknown if not provided); and need to continue or start medication for opioid use disorder (yes or no).

The Hub team also tracked the number of referrals received per month and the time between referral submission and release (“time to release”). Additionally, they determined location information, such as county of release and housing status at the time of referral. Housing status was categorized into 5 groups: residential address provided, transitional reentry housing address provided, only county provided as proxy, city or zip code provided as proxy, and parole/probation address provided as proxy address. Finally, the Hub staff noted the specific clinic to which each patient was referred.

### 2.4. Data Analysis

The Hub team analyzed program data from 8420 referrals received between April 2020 and August 2022. We removed 101 duplicate entries. We also tabulated and tracked the number of referrals from CDCR received by month across this period. We calculated the total number of hotline calls received by month during this same period and whether it was a pre-release or post-release call. The Hub team then summarized descriptive data on patient age, gender, and clinical needs. We determined the average time to release by month and analyzed changes in time to release over time. Fourteen patients did not have release dates listed and were excluded from this analysis.

We compiled referrals by county of release to determine the counties to which patients most often release to (top counties of release) and assessed for the proportion of all referrals in each housing status category at the time of referral.

Finally, we analyzed the catalogue of primary care resources to explore the attributes of clinics to which patients were referred for care following release. We tabulated the number of clinics affiliated with TCN versus not and those with providers willing to prescribe MOUD. We also explored secondary sources [41,42] to understand the community health resources available to patients in the counties to which they were returning home, including access to primary care providers (PCPs), specialty providers, and providers prescribing MOUD, compared to the state averages.

## 3. Results

### 3.1. Hub Baseline Data

The Hub completed 8420 referrals between its inception in April 2020 and August 2022. During the first year (2020), TCN received an average of 117 referrals per month, a number which increased to an average of 471 referrals per month for 2022. While there are monthly deviations in the number of referrals submitted to the Hub by CDCR, the system has been consistently utilized since its implementation, with the Hub receiving a high of 539 referrals in February 2022. See Figure 2 for referral trends by month. The Hub received 1524 calls on the reentry health care hotline during this same timeframe, with 540 (35%) of these calls coming from inside prisons or jails and 984 (65%) from callers recently released to the community. TCN provides resources to meet the needs of people with clinical needs across gender and age. See Table 2 for a summary of patient demographic distributions. See Table 3 for summary data on clinical needs.

### 3.2. Hub Referral Care Transitions Data

In addition to assessing the attributes of patients referred, we examined data related to care linkages.

#### 3.2.1. Housing Information

An analysis of housing status for patients referred revealed gaps in access to precise address information, which alludes to a problem with housing instability in this population. Of Hub referrals, 48.8% (4108) had a residential address provided, indicating the patient’s plan to live in their own home or with a spouse, family member, or friend following release. A small proportion of patients—8.2% (687)—had transitional reentry housing assigned to them at the time of referral. A total of 43% of patients (3625) had no specific address information at time of referral. Of these referrals lacking addresses, 37.5% (1360) provided only the county of release, 53.8% (1951) provided city or zip code information, and 8.7% (314) included a probation/parole address to use as a proxy (see Figure 3) A portion of referrals received by the Hub (134) was undetermined for the county of release, listing more than one potential county where a patient may be living and accessing services.

#### 3.2.2. Time to Release

For referrals submitted to TCN, only 38% (3173) were sent to the Hub with more than 30 days’ notice. Twenty-three percent of referrals (1881) were requested with less than 7 days to plan for release, with 6% of referrals (475) being submitted within 2 days of the patient’s release date (see Figure 4 for categories of time to release). Looking at monthly trends (Figure 5), the average time to release trended upwards overall between April 2020 and August 2022, from an average of 26 days in 2020 to an average of 34 days in 2022, with noticeable fluctuation throughout this whole timeframe. April and May 2020 stand out as outliers, with the average time to release starting at over 40 days, likely skewed by a low sample of referrals in these initial months of Hub implementation, when the COVID-19 pandemic was just emerging in the US (the combined number of referrals for April and May of 2020 was 40, which is merely 0.004% of the total referrals received). The number of incoming referrals to the Hub had spiked as of June 2020, after which the time to release began to trend upwards and continued, with variation, until August 2022.

#### 3.2.3. Resources across Communities of Release

The Hub’s catalogue of primary care resources included 436 different primary clinic sites across California. For 39.2% of the referrals received from resource nurses for patients requiring MOUD services in the community, the Hub referred these patients to 191 different primary care clinic sites across California with providers willing prescribe MOUD. Nineteen percent of these statewide clinic resources were affiliated with TCN. Based on the geographical proximity of patients to these clinics, the Hub was able to refer 4162 patients (49.4% of the total referrals) to the CHWs at these TCN-affiliated clinics, while the other 50.6% of patients were referred to clinics outside of TCN’s network because their location of release in the community was not proximate to a TCN site.

Figure 6 maps the ten top counties of release for patients referred to primary care through the Hub, representing 77% of this population. This map also indicates the location of California state prisons with yellow dots. Resources for primary care, specialty care, and SUD treatment vary drastically by county across the state. For these ten counties where most of the Hub’s patient population is releasing, there is a deficit in resources for primary care, specialty care, and SUD treatment, despite the high needs of the reentry population. Half of these counties have fewer PCPs, and four out of the ten top counties of release have fewer specialty care providers than the statewide averages. Eight out of the ten top counties of release have insufficient providers to provide MOUD to residents of that county with OUDs. For instance, Riverside County has 38% fewer PCPs and San Bernardino has 48% fewer specialists as the statewide average. San Bernardino and Kern Counties have the greatest disparities in providers offering MOUD despite higher rates of OUD. These three under-resourced counties represent 12% of patients (1026) referred to the Hub. See Table 4 for a summary of these county-level resources.

## 4. Discussion

This paper describes the Hub program to systematically improve care coordination between a statewide prison system and community-based primary care clinics. The Hub is a community-driven intervention has created a pathway for patient information to flow from behind the walls of state prisons directly into the hands of community-based health care professionals who assist patients with their reentry health needs. While there are published examples of other small-scale, local referral pathways, the Hub links chronically ill patients from a statewide prison system to statewide primary care resources. The program is responsive to the unique nature of releases from prisons, where patients are incarcerated further away from home, leaving incarceration after longer periods, and returning to communities with varying resources during a global pandemic. All patients referred through this system received timely information about health insurance, community primary care resources, social resources, and TCN’s safety-net reentry health care hotline. Key learnings from the Hub are that reentry care continuity requires time and access to patients for pre-release planning, the sharing of meaningful data, and robust patient-centered services in the community.

First, time to release and access to patients is challenging with incarceration and impacts care planning. Release dates from state prison may be known in advance or may be unanticipated and are also subject to change from internal prison policies and circumstances. Therefore, the window of time for which to plan for an upcoming prison release varies and is often limited. However, a short or variable timeframe of planning and limited channels to communicate with patients inside prisons prevents patient engagement and more comprehensive efforts to link patients to resources. There is evidence that post-release engagement in care is improved by pre-release interactions with community-based care providers [43]. Yet, routine in-person pre-release interactions were not feasible as part of this program due to COVID-19 and prison policies. Efforts to directly connect with people in prison through the hotline were limited and data show most calls came from the community after release. Prior to release, patients may have limited access to phones and may not have enough information to plan for release, so they make the call once in the community while navigating next steps in their care. The average time to release for Hub referrals improved as implementation of the program progressed, though fluctuation in the monthly trends highlights the challenges in planning for prison releases. Time to plan for release was likely additionally impacted by the COVID-19 pandemic, with illness and outbreaks contributing to expedited release dates, sporadic prison lockdowns, and staffing shortages. A limitation of these data is that the original release date submitted in the referral survey may not accurately represent when the patient left prison if changes occurred internally at the prison. Despite these challenges, when possible, a full 90 days of pre-release planning is recommended for primary care partners to assess patients and their needs, build relationships for successful engagement, and coordinate post-release care. When release dates can be anticipated, there is opportunity for earlier referral to community-based care providers and more engagement between carceral and community entities to support pre-release care planning, including leveraging telehealth tools to provide more access to incarcerated patients from outside prisons. When 90 days is not possible or when patient access is limited, there must be structures in place to prioritize coordination for the most complex patients, and robust safety-net systems need to be in place for patients upon return. These include our reentry health care hotline and enhanced community clinical programs such as TCN, which provides patient-facing real-time support and resources to individuals immediately after their release. Second, access to information is critical for making an accurate referral and sharing medical information to a specific clinic for continuity of care. Housing after release is an important component of successful reentry and a lack of accurate post-release housing information impedes making an effective referral to care. A patient’s housing status or even county of release may be undetermined at the time of referral for several reasons, such as pending transitional housing placement or a patient requesting transfer to a different county. Using a probation/parole address as a proxy assumes that the clinic closest to the probation/parole office is the most convenient, accessible, or preferred location for the patient. To address this challenge, the Hub referral coordinators often relied on providing general health care resources or having patients call TCN’s reentry hotline for assistance once they were released. The lack of these data presents challenges not only for linkage, but also for access to care. In California where Medicaid is county-based, last minute changes in county residence creates additional delays in insurance coverage. There is a need to collect more accurate data on housing status for effective linkages and to assess the impacts of housing instability for people releasing from prison on engagement in care post release. At minimum, there should be continuous communication between carceral and community entities who are involved in care planning as information changes or becomes available. Generally, data sharing is a persistent challenge when prisons, community clinics, and social service entities utilize different data systems. With the current referral survey, TCN has noted gaps in data collected about race/ethnicity, the length and type of incarceration, probation/parole status, insurance status, COVID status, and behavioral health needs—information that would improve the ability to identify the highest risk patients, ensure that they receive appropriate care upon release, and improve community care providers’ ability to serve complex patients. Data on race and ethnicity are particularly important given the disproportionate impacts of mass incarceration on communities of color [44,45]. Health information exchanges could enhance access to data between the many entities that comprise the carceral and community health systems.

Third, Hub data demonstrate that people leaving prison have significant comorbidities and disproportionate access to health care resources in communities of release. Where a patient is released has implications for the health care resources available to them post-release. Decisions within prisons concerning the timing and location of release are primarily driven by safety considerations, more so than patient preference, health care needs, and existing social supports [46]. People often return to under-resourced regions that vary in resources, so enhancing clinical services at existing primary care clinics is one way to address reentry health needs. Nearly half of patients referred to primary care through the Hub were referred to clinics implementing the TCN model of care however, which is a high proportion of total referrals compared to the rest of the clinics outside of this network. More programs that specialize in providing evidence-based services to returning community members are needed, especially in communities most impacted by incarceration. Specific investments to grow the capacity of primary care clinics to serve this population are needed. This would include funds for clinic systems to grow integrated behavioral health services, implement trauma-informed care practices, and integrate evidence-based programs such as TCN. Clinics with TCN programs employ CHWs with lived experience of incarceration. Investments in technical assistance and training are needed to change the health system’s discriminatory hiring policies away from excluding CHWs candidates with criminal records and to successfully integrate and supervise CHWs onto clinical teams. Coaching for system transformation is generally needed to ensure services are more patient-centered (such as improving access through open access or having more flexible late policies). Additional funds for clinics to purchase transportation resources (bus passes, taxi vouchers), technology coaching, food pantry items, work clothing, and over the counter medical supplies incentivize engagement and support basics needs.

Establishing linkages across large and disjointed systems is challenging, and the Hub has demonstrated that coordination between state prisons and a network of diverse community-based primary care clinics is possible. This required sustained relationship building to establish communication, building off pre-existing care linkage efforts and an existing statewide network of primary care clinics with patient-centered evidence-based programs. A human-centered design approach that included input from leadership, frontline staff, and CHWs with histories of incarceration allowed for the development of the Hub program and the ongoing implementation, as demonstrated by its integration across both prison institutions and TCN community clinics. Through the Hub, both parties work to meet shared goals, with each entity offering contributions and mutually benefiting from improved data sharing and better care connections. Prison systems benefit from this connection through access to accurate and timely information about community health care resources and an improved capacity to support patients with post-release navigation. Community health systems benefit by having information from the prison system about upcoming patient releases, improving a CHW’s ability to locate patients in the community and to anticipate their immediate needs.

There is an opportunity to continue to grow this connection, expanding upon what has been learned so far during the pandemic to address systemic failures. Since the Hub was developed during a global emergency, simple technology systems were put in place to quickly share information, emergency funds were leveraged, and access to patients was necessarily restricted. Now, investments in infrastructure that streamline communications, improve data tracking and quality improvement efforts across systems, and afford greater direct access to incarcerated patients are needed. Another key next step is to conduct more research on the outcomes of these referrals to assess the impact of this intervention for care linkages on engagement in care. A reasonable next step is to obtain qualitative input from key stakeholders to the program’s development (nurses, CHWs, leadership, and patients) on the impact of this system being in place. Finally, the pandemic required establishing a remote referral workflow; this referral process could be improved with interventions that more actively engage the patient and their families in planning and creating pre-release connections to the community. This engagement should include collecting data directly from patients, incorporating patient preferences about follow-up care, and increasing connections between patients and the community health system before they are released. The involvement of CHWs with shared lived experience of incarceration and reentry can improve patient engagement in health care, and more research is needed to quantify the number and type of pre-release interactions that best facilitate post-release engagement and improved health outcomes.

This is a program description of a specific California-based referral intervention, so findings from this paper cannot be generalized. The data discussed in this paper reflect only the population of individuals CDCR referred to TCN for care coordination and may not represent the whole of the prison population. Nonetheless, these findings are relevant for efforts to link prison systems to community health systems. In other states, where laws and policies differ from California, connections across community and carceral systems can still be created, even if they exist on a smaller scale [47]. Notably, the Centers for Medicare and Medicaid (CMS) recently issued guidance for a Reentry Section 1115 Demonstration Opportunity, which would allow for states to waive the Medicaid Inmate Exclusion policy and activate Medicaid 30 days pre-release. This proposed policy would support warm handoffs between carceral and community health systems aiming to improve health outcomes by funding care coordination from incarceration to the community through Medicaid. The Hub program demonstrates that it is possible for an independent community-based clinic system to collaborate to create a statewide network of service providers to support the complex medical and social needs of returning community members. Findings from the Hub can inform states’ planning in linking carceral and community health systems and guide investments in infrastructure and workforce. Investments in primary care evidence-based programs such as the TCN program, expanding primary care-based behavioral services and workforce development for CHWs with lived experience of incarceration, and quality care and data-sharing requirements for carceral systems are needed for the success of these systems. Currently, 14 states have applied for this waiver [48]. This work is particularly significant in California where a monumental reform of the state’s Medicaid system (Medi-Cal) is underway through California Advancing and Innovating Medicaid (CalAIM). The CalAIM Justice Involved Initiative aims to fund care coordination through Medi-Cal up to 90 days pre-release as well as robust community-based services through a benefit called Enhanced Care Management (ECM) [49]. To achieve this, it will be necessary to engage stakeholders at both ends of the spectrum, from carceral health care to community health and reentry providers. For as long as people are incarcerated, improving reentry health care in California and other states will require that prison systems provide access to community providers to facilitate care coordination and linkages to culturally responsive healthcare services. Interventions in both community health systems are needed to mitigate the challenges of care continuity at reentry and to improve health equity for this population, in order for these new 1115 demonstrations to be successful.

## 5. Conclusions

This paper examines the creation of a systematic program to improve access to incarcerated people with health needs by community-based primary care clinics with enhanced services across California. This referral pathway has addressed some existing challenges with care coordination across systems, such as the improved exchange of information and enhanced connection between carceral and community entities. It also identified components critical to care continuity for the reentry population, including data sharing between health systems, sufficient time and patient access for pre-release connections, and more reentry-focused primary care resources. Learnings from the Hub model can be useful for care providers and policy makers working with people who are incarcerated or recently released. Investments are needed to strengthen community health care resources and to promote collaboration between primary care systems statewide. These system-level changes, led by those most impacted by incarceration, aim to improve health equity for this population.

## Figures and Tables

**Figure 1 ijerph-20-05806-f001:**
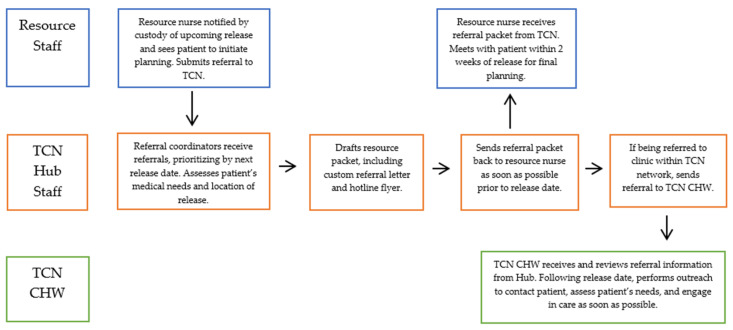
Visual representation of the referral workflow.

**Figure 2 ijerph-20-05806-f002:**
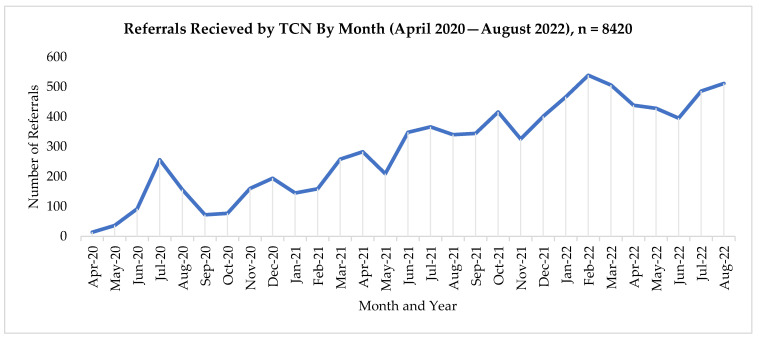
Referral trends by month.

**Figure 3 ijerph-20-05806-f003:**
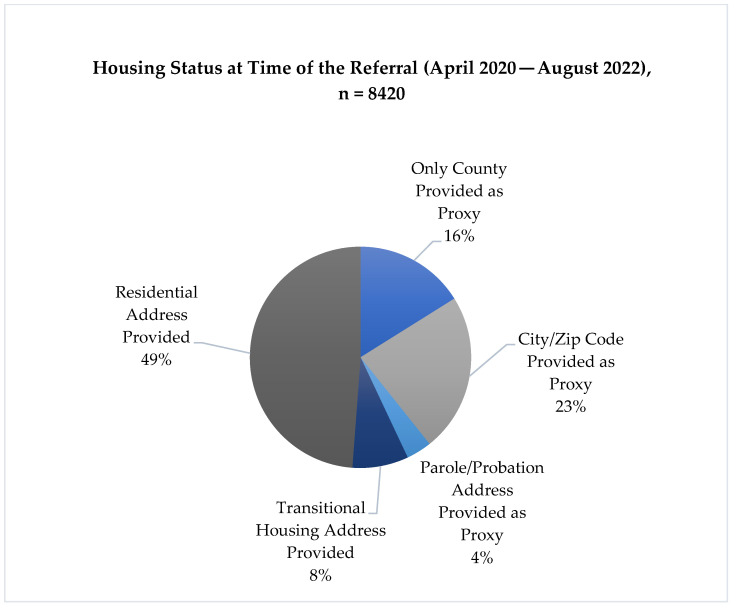
Patient housing status information at time of referral to TCN.

**Figure 4 ijerph-20-05806-f004:**
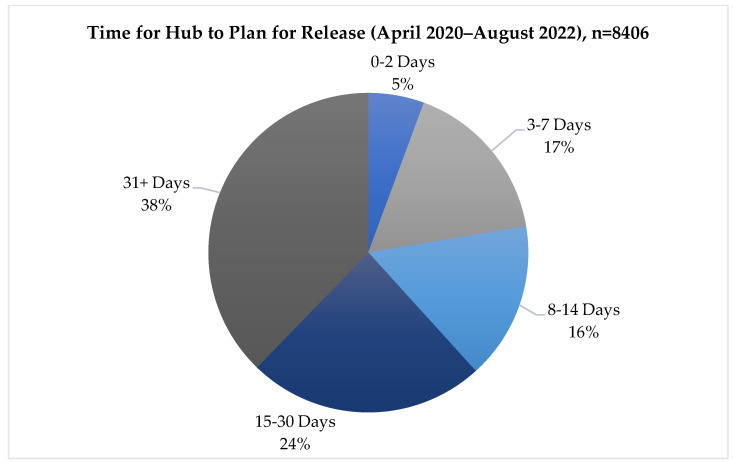
Time to release (overall).

**Figure 5 ijerph-20-05806-f005:**
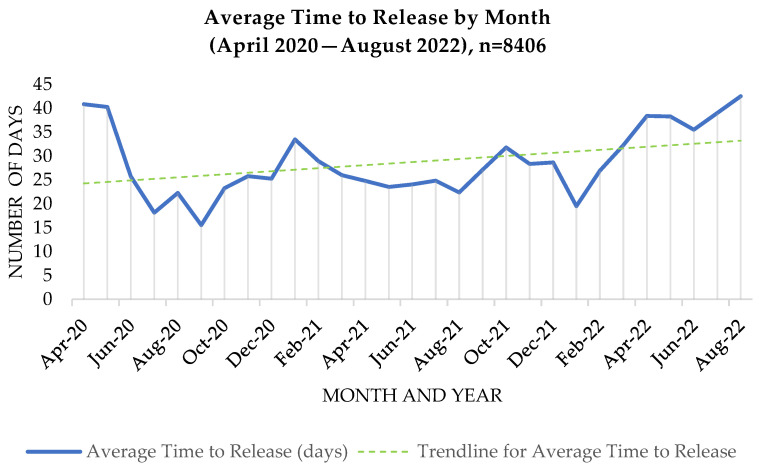
Time to release (by month).

**Figure 6 ijerph-20-05806-f006:**
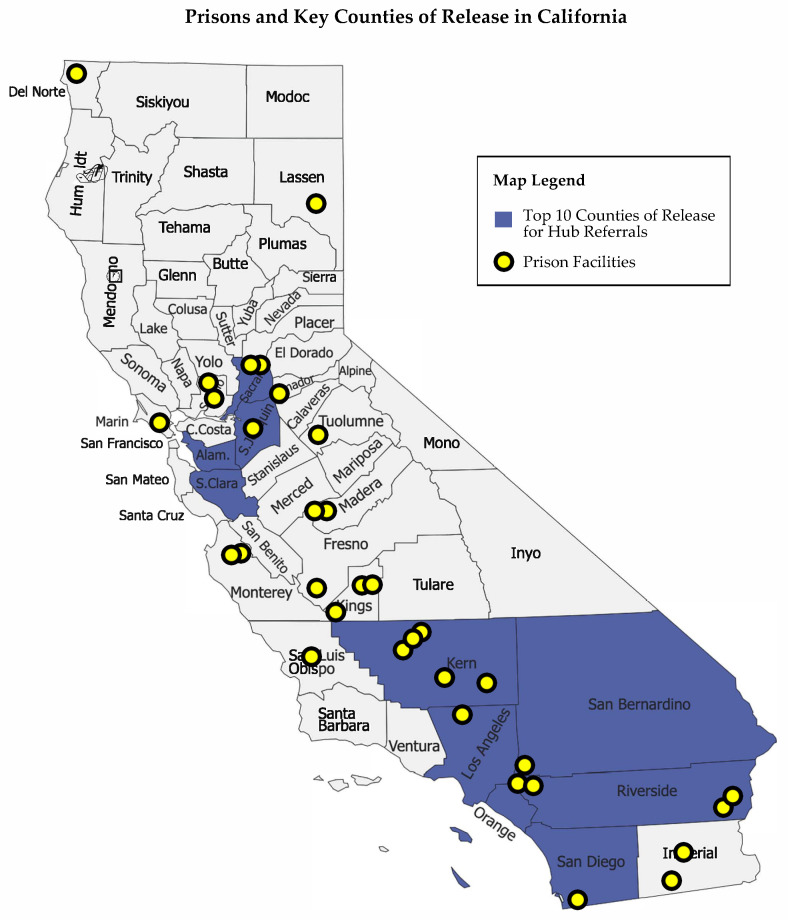
Map of California showing prison locations and the Hub population’s top counties of release.

**Table 1 ijerph-20-05806-t001:** Features of TCN’s Enhanced Primary Care Model.

CHW(s) with lived experience of incarceration are hired and integrated into primary care team. CHW leads patient outreach and engagement into care through building trusted relationships with patients and establishing diverse referral pathways. Primary care team provides integrated trauma-informed medical, mental health, and substance use treatment services. CHW supports patient with health and social service navigation, health education, cultural interpretation, and mentorship. Program serves patients through developing partnerships with community-based organizations, carceral entities (including prisons, jails, probation/parole), housing providers, legal services, and more. Program assesses and addresses social determinants of health related to reentry, including insurance eligibility, transportation, housing, employment, legal advocacy, food security, technology literacy, and family reunification. Program provides low-barrier ways to access and engage in care. Duration of case management services is not time-limited, rather based on the individual needs of each patient. CHWs and clinical team advocate for the needs of the reentry population to decision makers.

**Table 2 ijerph-20-05806-t002:** Demographic information of patients referred.

Characteristic	Category	Total	Proportion
Gender	Male	7963	94.5%
	Female	419	5.0%
	Transgender *	38	0.5%
Age	Age 19–21	24	0.3%
	Age 22–24	242	2.9%
	Age 25–34	2625	31.2%
	Age 35–44	2573	30.6%
	Age 45–54	1438	17.1%
	Age 55–64	1073	12.7%
	Age 65–74	380	4.5%
	Age 75–84	58	0.7%
	Age 85+	7	0.1%

***** Tracking for the “Transgender” category began in January 2022, so this is undercounted.

**Table 3 ijerph-20-05806-t003:** Clinical needs ***** of patients referred.

Characteristic	Category	Total	Proportion
Medical Risk	High	710	15.9%
	Medium	2048	45.7%
	Low	1338	29.9%
	Unknown	383	8.5%
Mental Health Risk	Acute (Inpatient/Crisis Bed)	61	1.4%
	Severe and Persistent (Enhanced Outpatient Program)	428	9.6%
	Mild–Moderate (Case Management)	1295	28.9%
	None	2291	51.1%
	Unknown	404	9.0%
MOUD Need	None (Primary Care only)	2706	60.4%
	Primary Care and MOUD	1692	37.8%
	MOUD only	81	1.8%

***** Tracking for clinical needs began in September 2021, so these data represent referrals from September 2021 to August 2022.

**Table 4 ijerph-20-05806-t004:** Health care services available in the Hub population’s ten top counties of release.

County of Release	Proportion of Hub Referrals	PCPs per 100,000 (State Average 49.8 per 100,000)	Specialists per 100,000 (State Average 104.6 per 100,000)	Average Proportion of Providers who Prescribe MOUD (State Average 3.2% of Providers)
Los Angeles	29.38%	48.1	110.3	2.6%
San Diego	9.23%	51.4	116	2.7%
Orange	7.92%	52	108.4	2.5%
Sacramento	7.00%	50.3	122.3	3.4%
San Bernardino	4.93%	38.3	47.6	2.0%
Riverside	4.63%	31	55.8	2.9%
San Joaquin	4.41%	39	65.4	2.8%
Alameda	3.67%	60.9	117.2	3.9%
Santa Clara	3.03%	65.6	149.1	2.5%
Kern	2.62%	36.4	55.8	1.9%

## Data Availability

The data presented in this study are available on request from the corresponding author. The data are not publicly available due to privacy.

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
