# Peer review of "The Reentry Health Care Hub: Creating a California-Based Referral System to Link Chronically Ill People Leaving Prison to Primary Care"

_ijerph, 2023, doi:10.3390/ijerph20105806_

Round 1

Reviewer 1 Report

Article: The Reentry Health Care Hub: Creating a California-based referral system to link chronically ill people leaving prison to primary care

Journal: International Journal of Environmental Research and Public Health

Authors: Bethany Divakaran, Natania Bloch, Mahima Sinha, Anna Steiner, and Shira Shavit

Overall Comment: The authors touch on an incredibly important topic among a vulnerable and understudied population (improving continuity of care among individuals recently released from prison), and I commend the work they have done here. While I believe there’s a lot of value in the authors’ work and believe it could make a significant contribution to the literature, the manuscript would benefit from significant revision. Overall, there’s a lot of vague and clunky language used throughout that makes the manuscript challenging to follow. Also, in some instances, particularly the methods and results section, the manuscript does not follow organizational practices one would expect in scientific/academic writing. I highly recommend the authors utilize an editing service to assist with organization, flow, sentence structure, and word choice within the manuscript. If not available through the journal, universities often provide resources to cover such services. I also recommend finding model papers in the literature (e.g., IJERPH) to use as a guide for structure. Note, I focused my review on more major comments, but there were many minor comments that I couldn’t capture here but would easily be addressed by an editing service. But in general, more clear, concise, and explicit language would be beneficial. Active (rather than passive) voice would also help. I have provided several suggestions/comments for the authors to consider that may help strengthen the paper.

Specific Comments:

Introduction

1.     There’s a couple of places where the authors use words of comparison (i.e., more, greater, higher than) but do not explicitly state the comparison they’re making. Please clarify who/what you are comparing (e.g., reentry pop vs. general pop)

a.     In line 35, the authors state, “The reentry population experiences more emergency room visits and hospitalization…” More than who/more than what? More than the general population?

b.     In line 37, the authors state, “This population has 12 times greater risk of death in the two weeks following release…” Greater risk than who/what?

c.      In line 38, the authors state, “…a 129 times higher risk of death by overdose…” higher risk than/compared to what?

2.     Lines 57-58, the author’s state, “Community clinics often will not schedule appointments for patients without active insurance.” While this is certainly true for some clinics, I would be careful not to overstate this without qualifying it or providing citations/evidence to support it, especially given that the purpose/mission of most community health centers and community clinics is to serve the underserved, including uninsured patients.

3.     Line 81, the authors state, “Most of these primary health systems are not part of county-level health services and operate independent of each other.” The authors should explain the relevance of this observation. It’s not clear why this is important. What problem does this pose as it relates to the focus of your paper?

4.     Line 82, the authors state, “While state prison systems may share health care information with county entities, the fragmented nature of this large primary health care system makes it difficult to exchange patient data with individual clinics.” However, the authors already stated in the previous sentence that most of the health systems are not county entities, so the relevance of this statement is unclear. Also, what about it makes it fragmented and how’s that related to the exchange of information?

5.     Lines 147-48, the authors state, “Here, we describe a first-of-its-kind statewide program focused on health linkage…” I recommend restating the name of the program (when you say “statewide program”) so the reader knows you are talking about the Hub mentioned above. This will provide more of a connection/improve flow between the previous paragraph and this one.

Materials and Methods

6.     Line 164, name the survey platform (e.g., online survey via REDCap).

7.     Line 164-65, the authors state, “Resource nurses use this survey to submit information for patients with upcoming prison release dates…” Clarify who this information is submitted to. I assume it’s to partner clinics, but this should be explicitly stated. Also, how was it decided which clinic to submit a particular person’s information to (proximity to last known address)? I realize now this information is presented in a different section later, but this information should be presented together. Reconsider how best to present this information so the reader doesn’t have so many questions before they get to the answer.

8.     Line 178-79, the authors state, “TCN-affiliated clinics committed to receiving referrals from the Hub for patients recently released from prison, with the CHW being the first point of contact.” This information should be stated before mention of the referral platform since these are the entities receiving the referral information.

9.     Lines 180-82, the authors mention the hotline but it’s not clear who/what the hotline is for. I see that you describe this in a later section, but again, this information should either be clarified here or moved to where you describe it later.

10.  Lines 190-91, the authors state, “These clinics may be within the TCN network of community primary care clinics or outside of it, depending on the patient’s community location and preference.” Here the reader is left wondering what happens when it is outside of the network. I recognize it’s described later, but again, either explain here or don’t bring it up here.

11.  Line 206, the authors state, “Hub staff may have assisted resource nurses…” Seems odd to say, “may have.” Are the authors unsure? I would make this a declarative statement.

12.  Line 209, the authors state, “TCN CHWs engaged in outreach to locate patients in the community.” What do you mean by they located patients in the community? The framing is odd. It makes it seem like the CHWs were looking for patients, maybe to recruit. Do the authors mean something like, “CHWs contacted patients once they were released from prison?” Please clarify.

13.  Figure 1:

a.     First blue box – Indicate who notifies the resource nurses.

b.     Second blue box – Clarify who the resource nurse is coordinating with and who the patient is meeting with.

14.  Overall, the data collection and analysis section would benefit from additional detail being presented and in a more structured way. Typically, data variables are described first, then how the data were manipulated/analyzed is described. I recommend looking to other published articles, in particular within IJERPH, as a model to guide the reworking of this section. For example:

c.      Medical risk score, mental health risk score, SUD, and other special needs are not adequately described/defined. Please describe the meaning and/or criteria used to calculate each variable.

d.     “Medical risk score” is used in this section but “physical health” is used in the results section. The authors should be consistent in their terminology throughout.

e.     Please indicate what the categories are for each of your categorical variables (e.g., gender, age, address, risk scores). This should be described in the methods before we see it in the results.

f.       I’m curious whether data on chronic disease was collected since this program is for patients with chronic disease. Is it captured in one of these risk scores? If so, this should be indicated in the definition of the relevant risk score.

g.     There are multiple variables described in your analyses portion of this section that were not first defined in the data collection portion of this section (e.g., number of referrals, time to release, average time to release). All variables should be defined before describing how they were manipulated/analyzed.

15.  Line 233, “analyzed for the source of the call” is awkwardly phrased/unclear so it was difficult to understand. It was not until I got to the results section that I understood what the authors meant here. I recommend reframing. Something like the following may be clearer, “We assessed the total number of hotline calls received and whether it was a pre-release or post-release call.”

Results

16.  Line 270 is the first time anything about co-morbidity comes up. If there is a measure of co-morbidity, it should be indicated in the methods. 

17.  Lines 269-71, is awkwardly positioned between two tables, which makes it easy to miss and appear as it is part of one of the tables. I recommend either creating more space around this text to make it clear it is its own paragraph or moving it up to be included in the first paragraph. The latter would be the preferred option because there really isn’t enough information here to warrant its own paragraph (technically, a paragraph must have at least 3 sentences).

18.  Table 2 is titled “Clinical Need and Co-Morbid Conditions of Patients Referred,” however, the information presented in the table does not speak to co-morbidity. It only appears to list various health statuses. Are these scores in some way related to co-morbidities? If so, that should be described in the methods section.

19.  It’s not clear what “high, medium, low” physical health means. As mentioned previously, the risk scores and their categories should be defined in the methods section.

20.  Line 279, clarify what is meant by “waivered.”

21.  The authors frequently include information in the results section that would be more appropriate in the methods and/or discussion sections. Definitions, background/rationales, and descriptions of how or why they did something should be in the methods. Interpretations of the results (besides maybe minor comments/observations) should largely be in the discussion section. For example:

a.     Table 3 describes the model used which should be in the methods. It’s not a result.

b.     The information in lines 290-97, 299-301, 304-05 should be in the methods and/or discussion.

c.      There shouldn’t be citations in the results section (re: lines 346-47). If these are the sources reviewed to compile this information, it should be reported in the methods section.

22.  Figure 4: Consider combining these two pie charts into a single chart with 1) Residential Address Provided, 2) Transitional Housing Address Provided, 3) Parole/Probation Address Provided as Proxy, 4) City/Zip Code Provided as Proxy, and 5) Only County Provided as Proxy.

23.  Statements like “best” and “better” are used in lines 352-53. These are qualitative statements, which are not appropriate without support (which would then likely go in the discussion). “Best” based on what? If by “best,” the authors mean “has the most resources,” then the latter is how this should be reported in the results section.

Discussion

24.  Lines 397-99, the authors recommend a minimum of 90-days of pre-release planning. However, earlier in the paragraph, the authors provide valid reasons for why advanced notice of release is sometimes not feasible. So, what’s the recommendation for those instances?

25.  Lines 427-30, the authors recommend “building robust primary health care systems in the community that are responsive to the demands of reentry” as a remedy for the lack of resources available in the communities people return to. Of course this is true, but this is too general for a recommendation and there are real challenges in these neighborhoods keeping this from happening so just leaving it at that isn’t sufficient (and may come across a bit naïve/disconnected). It would be helpful to have other more concrete and feasible recommendations as well. What's within the means of those who may look to this paper for guidance on implementing their own reentry program. They're almost certainly not going to create a new primary health care system, so what else can they do? Consider who your audience is (this may be multiple sectors) and speak to what they can/should do.

26.  Lines 449-59, the authors make several broad claims about “successful implementation,” describing outcomes that haven’t been backed up in the paper with data. The authors should be careful to only draw conclusions based on the results that were actually presented, unless they have some other support from the literature.

Author Response

Response to Reviewer 1 Comments

Overall Comment: The authors touch on an incredibly important topic among a vulnerable and understudied population (improving continuity of care among individuals recently released from prison), and I commend the work they have done here. While I believe there’s a lot of value in the authors’ work and believe it could make a significant contribution to the literature, the manuscript would benefit from significant revision. Overall, there’s a lot of vague and clunky language used throughout that makes the manuscript challenging to follow. Also, in some instances, particularly the methods and results section, the manuscript does not follow organizational practices one would expect in scientific/academic writing. I highly recommend the authors utilize an editing service to assist with organization, flow, sentence structure, and word choice within the manuscript. If not available through the journal, universities often provide resources to cover such services. I also recommend finding model papers in the literature (e.g., IJERPH) to use as a guide for structure. Note, I focused my review on more major comments, but there were many minor comments that I couldn’t capture here but would easily be addressed by an editing service. But in general, more clear, concise, and explicit language would be beneficial. Active (rather than passive) voice would also help. I have provided several suggestions/comments for the authors to consider that may help strengthen the paper.

Response: Our team has made significant revisions to this paper, particularly concerning organization of each section of the paper in alignment with standards of academic manuscripts. While we defer the use of an external editing service at this time, we have referred to other IJERPH papers for additional guidance on structure. Further, we have made edits to address issues of clarity and voice throughout the paper. Please see more specific revisions below.

Introduction

  1. 1.There’s a couple of places where the authors use words of comparison (i.e., more, greater, higher than) but do not explicitly state the comparison they’re making. Please clarify who/what you are comparing (e.g., reentry pop vs. general pop)
  2. In line 35, the authors state, “The reentry population experiences more emergency room visits and hospitalization…” More than who/more than what? More than the general population?

        Response: Added clarification “than the general population”.

  1. In line 37, the authors state, “This population has 12 times greater risk of death in the two weeks following release…” Greater risk than who/what?

        Response: Added clarification “than the general population”.

  1. In line 38, the authors state, “…a 129 times higher risk of death by overdose…” higher risk than/compared to what?

        Response: The earlier part of the sentence specifies the comparison is to the general population.

  1. Lines 57-58, the author’s state, “Community clinics often will not schedule appointments for patients without active insurance.” While this is certainly true for some clinics, I would be careful not to overstate this without qualifying it or providing citations/evidence to support it, especially given that the purpose/mission of most community health centers and community clinics is to serve the underserved, including uninsured patients.

        Response: While we have experienced barriers at some clinics when trying to schedule appts for patients who are still incarcerated, we agree we do not want to over-generalize. We restate here “a gap in insurance upon reentry may be a barrier”, rather than stating “clinics often will not schedule”.

  1. Line 81, the authors state, “Most of these primary health systems are not part of county-level health services and operate independent of each other.” The authors should explain the relevance of this observation. It’s not clear why this is important. What problem does this pose as it relates to the focus of your paper?

        Response: Given we are not focusing on county-level service provision and relations in this paper, we removed reference to county health services here, to simply make the point that in the community FQHCs are disjointed from one another.

  1. Line 82, the authors state, “While state prison systems may share health care information with county entities, the fragmented nature of this large primary health care system makes it difficult to exchange patient data with individual clinics.” However, the authors already stated in the previous sentence that most of the health systems are not county entities, so the relevance of this statement is unclear. Also, what about it makes it fragmented and how’s that related to the exchange of information?

        Response: Removing the county reference and focusing on the attributes of community-based clinics and the way information is shared between prisons and clinics, restated as “Most of these primary health systems operate independent of each other and are not coordinated in providing services or sharing patient information, the result being that patients may receive medical services from more than one disjointed systems. Community clinics do not routinely receive information from the prison system for patients returning to the community.”

  1. Lines 147-48, the authors state, “Here, we describe a first-of-its-kind statewide program focused on health linkage…” I recommend restating the name of the program (when you say “statewide program”) so the reader knows you are talking about the Hub mentioned above. This will provide more of a connection/improve flow between the previous paragraph and this one.

        Response: Referenced the Hub by name

Methods

  1. Line 164, name the survey platform (e.g., online survey via REDCap).

        Response: Specified platform (Qualtrics XM)

  1. Line 164-65, the authors state, “Resource nurses use this survey to submit information for patients with upcoming prison release dates…” Clarify who this information is submitted to. I assume it’s to partner clinics, but this should be explicitly stated. Also, how was it decided which clinic to submit a particular person’s information to (proximity to last known address)? I realize now this information is presented in a different section later, but this information should be presented together. Reconsider how best to present this information so the reader doesn’t have so many questions before they get to the answer.

        Response: Removed the sentence to which this comment is applying, since this section is discussing how the Hub formed. We combined the description of collaborating with CDCR and community-clinics into this one paragraph. The following section (2.2) describes the Hub referral workflow in detail. Here it is specified that referral information is sent to the Hub, then the Hub determines which community-based clinic to send patient’s information to, based on patient need and geographic location in the community.

  1. Line 178-79, the authors state, “TCN-affiliated clinics committed to receiving referrals from the Hub for patients recently released from prison, with the CHW being the first point of contact.” This information should be stated before mention of the referral platform since these are the entities receiving the referral information.

        Response: Described collaborative planning first in Section 2.1, then described the Hub intervention in Section 2.2.

  1. Lines 180-82, the authors mention the hotline but it’s not clear who/what the hotline is for. I see that you describe this in a later section, but again, this information should either be clarified here or moved to where you describe it later.

        Response: Moved sentence from Section 2.2 to Section 2.1, describing the hotline earlier so we can just reference it later.

  1. Lines 190-91, the authors state, “These clinics may be within the TCN network of community primary care clinics or outside of it, depending on the patient’s community location and preference.” Here the reader is left wondering what happens when it is outside of the network. I recognize it’s described later, but again, either explain here or don’t bring it up here.

        Response: We can remove this sentence here, since the issue is addressed in the next paragraph, thus answering the reviewer’s question.

  1. Line 206, the authors state, “Hub staff may have assisted resource nurses…” Seems odd to say, “may have.” Are the authors unsure? I would make this a declarative statement.

        Response: Made this sentence more definitive, specifying the prison resource nurses schedule appointments and Hub staff support with coordination as needed. Also, described scheduling follow-up appointments and coordinating pre-release phone connections separately, for clarity.

  1. 12.  Line 209, the authors state, “TCN CHWs engaged in outreach to locate patients in the community.” What do you mean by they located patients in the community? The framing is odd. It makes it seem like the CHWs were looking for patients, maybe to recruit. Do the authors mean something like, “CHWs contacted patients once they were released from prison?” Please clarify.

        Response: This phrase “locating patients” is accurate in the sense that patients returning from incarceration can often be hard to contact and engage in the community. TCN’s CHWs sometime are “looking for patients” in the community - at transitional houses, parole meetings, etc. However, we don’t want it to come across as confusing to our readers, so we edited to clarify, stating “CHWs reached out to patients through available means (if they have contact information, trying to locate at parole meetings or transitional houses) to attempt to engage them in care”. 

  1. Figure 1:
  2. First blue box – Indicate who notifies the resource nurses.

        Response: For clarity, we add “notified by custody” to first blue box.

  1. Second blue box – Clarify who the resource nurse is coordinating with and who the patient is meeting with.

        Response: For clarity, we removed “Coordinates to meet” just simply state “Meets with patient…”.

  1. Overall, the data collection and analysis section would benefit from additional detail being presented and in a more structured way. Typically, data variables are described first, then how the data were manipulated/analyzed is described. I recommend looking to other published articles, in particular within IJERPH, as a model to guide the reworking of this section. For example:

        Response: Divided this part into two sections – 2.3 Data Collection & 2.4 Data Analysis

  1. Medical risk score, mental health risk score, SUD, and other special needs are not adequately described/defined. Please describe the meaning and/or criteria used to calculate each variable.
  2. “Medical risk score” is used in this section but “physical health” is used in the results section. The authors should be consistent in their terminology throughout.
  3. Please indicate what the categories are for each of your categorical variables (e.g., gender, age, address, risk scores). This should be described in the methods before we see it in the results.

Response: We defined all categories and categorical variables in Section 2.3 Data Collection. All terminology in Section 2.3 now aligns with characteristics and categories in Table 1 and Table 2 in results section (so rather than stating “physical health” we consistently use the term “medical risk”).

  1. I’m curious whether data on chronic disease was collected since this program is for patients with chronic disease. Is it captured in one of these risk scores? If so, this should be indicated in the definition of the relevant risk score.

Response: Chronic disease diagnosis is factored into CDCR’s medical risk. The medical risk characteristic is sorted into High, Medium, and Low categories “depending on the severity and complexity of chronic medical conditions or Unknown if not provided”. 

  1. There are multiple variables described in your analyses portion of this section that were not first defined in the data collection portion of this section (e.g., number of referrals, time to release, average time to release). All variables should be defined before describing how they were manipulated/analyzed.

Response: We moved all definitions of variables to Methods Section 2.3 data collection

  1. Line 233, “analyzed for the source of the call” is awkwardly phrased/unclear so it was difficult to understand. It was not until I got to the results section that I understood what the authors meant here. I recommend reframing. Something like the following may be clearer, “We assessed the total number of hotline calls received and whether it was a pre-release or post-release call.”

        Response: Corrected as suggested for clarity.

Results

  1. Line 270 is the first time anything about co-morbidity comes up. If there is a measure of co-morbidity, it should be indicated in the methods.

        Response: We removed the term “co-morbidities” as it is not a helpful description here. As defined in Methods Section 2.3 Data Collection, the term “clinical needs” is consistently used.

  1. Lines 269-71, is awkwardly positioned between two tables, which makes it easy to miss and appear as it is part of one of the tables. I recommend either creating more space around this text to make it clear it is its own paragraph or moving it up to be included in the first paragraph. The latter would be the preferred option because there really isn’t enough information here to warrant its own paragraph (technically, a paragraph must have at least 3 sentences).

        Response: As suggested, we moved some of this text to previous paragraph and some to footnote of “Clinical Needs” table.

  1. Table 2 is titled “Clinical Need and Co-Morbid Conditions of Patients Referred,” however, the information presented in the table does not speak to co-morbidity. It only appears to list various health statuses. Are these scores in some way related to co-morbidities? If so, that should be described in the methods section.

        Response: We removed the term “co-morbidities” as it is not a helpful description here. As defined in Methods Section 2.3 Data Collection, this table now refers to “clinical needs”.

  1. It’s not clear what “high, medium, low” physical health means. As mentioned previously, the risk scores and their categories should be defined in the methods section.

        Response: Medical risk and associated categories are now defined in Methods Section 2.3 Data Collection.

  1. Line 279, clarify what is meant by “waivered.”

        Response: We have removed the term “waivered” for clarity/simplification. Waivered in this context means providers are willing and able to prescribe MOUD; however, this is now an irrelevant qualification since a waiver is no longer required for physicians to prescribe MOUD.

  1. The authors frequently include information in the results section that would be more appropriate in the methods and/or discussion sections. Definitions, background/rationales, and descriptions of how or why they did something should be in the methods. Interpretations of the results (besides maybe minor comments/observations) should largely be in the discussion section.

        Response: We reassessed the structure of these sections so that the results section includes only objective results. See below for specific comments about examples noted:

  1. Table 3 describes the model used which should be in the methods. It’s not a result.

        Response: This Table showing “Features of TCN’s Enhanced Primary Care Model” has been moved to Methods Section 2.3 Data Collection Section. It is now Table 1.

  1. The information in lines 290-97, 299-301, 304-05 should be in the methods and/or discussion.

Response: Former lines 290-97 moved to Methods section. Former lines 299-301 deleted. Former lines 304-305 moved to Discussion section.

  1. There shouldn’t be citations in the results section (re: lines 346-47). If these are the sources reviewed to compile this information, it should be reported in the methods section.

        Response: These citations are now referenced in Methods Section 2.3 instead.

  1. Figure 4: Consider combining these two pie charts into a single chart with 1) Residential Address Provided, 2) Transitional Housing Address Provided, 3) Parole/Probation Address Provided as Proxy, 4) City/Zip Code Provided as Proxy, and 5) Only County Provided as Proxy.

        Response: Our methods section now defines “housing status at time of referral” by five categories: residential address provided, transitional reentry housing address provided, county only provided (the address line in the referral survey being blank or marked as “homeless”, “transient”, “unknown”, “n/a”, or “pending”), city or zip code only provided, and parole/probation address listed as proxy address. We reformatted this pie chart to include these five categories into a single visualization.

  1. Statements like “best” and “better” are used in lines 352-53. These are qualitative statements, which are not appropriate without support (which would then likely go in the discussion). “Best” based on what? If by “best,” the authors mean “has the most resources,” then the latter is how this should be reported in the results section.

        Response: We removed these subjective qualifiers to focus on what the data objectively states.

Discussion

  1. Lines 397-99, the authors recommend a minimum of 90-days of pre-release planning. However, earlier in the paragraph, the authors provide valid reasons for why advanced notice of release is sometimes not feasible. So, what’s the recommendation for those instances?

        Response: We acknowledge that 90 days for care coordination is not always possible. When possible, “there is opportunity for earlier referral to community-based care providers and more engagement between carceral and community entities to support pre-release care planning, including leveraging telehealth tools to provide more access to incarcerated patients from outside prisons.” When 90-days pre-release care coordination is not possible, we recommend putting structures in place to support care coordination beyond ISUDT, such as the hotline that the Hub team runs or advanced care management (which exists in TCN-affiliated clinics).

  1. Lines 427-30, the authors recommend “building robust primary health care systems in the community that are responsive to the demands of reentry” as a remedy for the lack of resources available in the communities people return to. Of course this is true, but this is too general for a recommendation and there are real challenges in these neighborhoods keeping this from happening so just leaving it at that isn’t sufficient (and may come across a bit naïve/disconnected). It would be helpful to have other more concrete and feasible recommendations as well. What's within the means of those who may look to this paper for guidance on implementing their own reentry program. They're almost certainly not going to create a new primary health care system, so what else can they do? Consider who your audience is (this may be multiple sectors) and speak to what they can/should do.

        Response: Health systems can transform their own practices and policies to better serve returning community members, including changing hiring practices to be more inclusive of staff with lived history of incarceration and shifting policies to be more patient-focused; increased investments in culturally responsive primary care from states or health plans; and providing safety net assistance, such as transportation, food support, etc.  

  1. Lines 449-59, the authors make several broad claims about “successful implementation,” describing outcomes that haven’t been backed up in the paper with data. The authors should be careful to only draw conclusions based on the results that were actually presented, unless they have some other support from the literature.

        Response: We removed such broad claims, given we do not yet have specific outcome data on certain measures of success. We are focusing here on the process outcome of having developed and put into a place a collaborative system that is still operational after 3 years.

Reviewer 2 Report

Manuscript Review - IJERPH-2209224

The Reentry Health Care Hub: Creating a California-Based Referral

System to Link Chronically Ill People Leaving Prison to Primary Care

This manuscript considers the important role of a community-based non-profit organization, the Transitions Clinic Network, with collaboration with the CDCR to establish the Reentry Health Care Hub, which supports direct linkages to care post-release for people being released from prisons in California. The manuscript is well-written and interesting (and important!) and I only have a few suggestions for improvement.

Introduction & Literature Review:

·         The authors may want to check on the Medicaid federal policy issue they bring up on line 54. I only say this because I believe, recently, they’ve changed some of the rules with that (at least certain states have), where someone is only “suspended” while they are incarcerated, rather than having to completely start anew again with Medicaid once they’re released.

·        The TCN and surrounding affiliated clinics in other states that it has built is quite impressive, and badly needed.

Methods & Results:

·         It is not clear, until line 222, that this paper is an evaluation of the Hub. Up until this point, it reads as if the paper will be merely descriptive. I highly suggest adding more clear information in the front end (and perhaps the title?) to indicate this is a program evaluation.

·         I think it’s important in section 2.3 to clearly list how many referrals or patients you started with before mentioning things like “we removed 101 duplicate entries” (line 223) or “Fourteen patients did not have release dates and were excluded…” (line 231). The reader has no idea how much missingness is here without knowing how many referrals and patients you started with.

·         What was the reason for Hub referral coordinators referring half of the patients to TCN-affiliated clinics and the others to clinics outside of TCN’s network? This should be added for clarity.

·         You mention in line 323 that April and May 2020 stand out due to being the initial implementation months, but I would also add here that the pandemic was in its infancy in America, with a lot of confusion during that time plus people were probably not being released for medical reasons until at least a little while after that.

·         I might have spoken too soon regarding this as an official evaluation. While the descriptive nature of the paper is important and interesting, I highly suggest some more pointed information that can better grasp how helpful the network is. Merely being descriptive at this point makes it really hard to understand what the full benefit is of the program. Authors should perhaps get creative and consider other ways to express what is beneficial about the program and how have things been since then (August 2022)? Maybe that is a better way to show how helpful the program is. Additionally, please remove verbiage related to “evaluation”, as this is not a true evaluation.

Discussion:

·         I believe the discussion is missing key information on how something like the Hub can be used in other states in similar situations. More specificity on this point is warranted.

·         What has changed with how the Hub deals with patients now that the pandemic is somewhat behind us? A bit more on “where we go from here” both in other states as well as post-pandemic.

·         Additionally, will the authors consider a more directed, qualitative analysis with the CHWs and/or other partners and/or patients who were critical in getting this off the ground? That seems to be the next, and clear, area of research to better establish how well the program did or did not work. This would be especially helpful for other DOCs wanting to partner with clinics in their states.

Minor Edits:

·         Line 262- I think the word “of” is missing between “summary” and “patient”

·         Line 276 & 308- I don’t think you can/should start the sentence with a number.

·         Appears as if reference 20 and 23 are the same. Same with 11 and 14 and 19. I believe you are only supposed to use the number once when referring to the same reference.

·         The thick lines between features of the TCN model (Table 3) are distracting, making the table hard to read. I suggesting using lightly dotted lines or listing them as a bulleted list for clarity.

·         The term PCP, listed in Table 4, is never spelled out/explained. Also, what is “X-waivered”? This is also never written out or explained. Both are in also in Table 4. More information on this table in general is warranted.

Author Response

Response to Reviewer 2 Comments

Overall Comments: This manuscript considers the important role of a community-based non-profit organization, the Transitions Clinic Network, with collaboration with the CDCR to establish the Reentry Health Care Hub, which supports direct linkages to care post-release for people being released from prisons in California. The manuscript is well-written and interesting (and important!) and I only have a few suggestions for improvement.

Introduction & Literature Review:

  1. The authors may want to check on the Medicaid federal policy issue they bring up on line 54. I only say this because I believe, recently, they’ve changed some of the rules with that (at least certain states have), where someone is only “suspended” while they are incarcerated, rather than having to completely start anew again with Medicaid once they’re released.

Response: The federal inmate Medicaid exclusion makes it the responsibility of state/local entities to pay for health care inside carceral settings, but it varies by state as to how this is implemented. Suspension vs. termination is determined by states, based on the time one is incarcerated (some states will suspend coverage for <1 year in jail rather than terminate immediately). In writing “suspended or terminated” we were aiming to cover both scenarios that may exist in different states. Made minor revisions to be clearer: “Due to federal policy, people who are incarcerated are excluded from Medicaid coverage. Depending on the state in which one lives, insurance coverage through Medicaid is suspended or terminated when one is incarcerated, and patients cannot reactivate Medicaid until back in the community.”

Methods & Results:

  1. It is not clear, until line 222, that this paper is an evaluation of the Hub. Up until this point, it reads as if the paper will be merely descriptive. I highly suggest adding more clear information in the front end (and perhaps the title?) to indicate this is a program evaluation… I might have spoken too soon regarding this as an official evaluation. While the descriptive nature of the paper is important and interesting, I highly suggest some more pointed information that can better grasp how helpful the network is. Merely being descriptive at this point makes it really hard to understand what the full benefit is of the program. Authors should perhaps get creative and consider other ways to express what is beneficial about the program and how have things been since then (August 2022)? Maybe that is a better way to show how helpful the program is. Additionally, please remove verbiage related to “evaluation”, as this is not a true evaluation.

Response: References to evaluation are removed, it is described as a “program description”. I feel like we’ve mentioned many examples of what it beneficial about this program, even though we don’t have specific outcomes data yet.

  1. I think it’s important in section 2.3 to clearly list how many referrals or patients you started with before mentioning things like “we removed 101 duplicate entries” (line 223) or “Fourteen patients did not have release dates and were excluded…” (line 231). The reader has no idea how much missingness is here without knowing how many referrals and patients you started with.

Response: Added total number of referrals - 8420

  1. What was the reason for Hub referral coordinators referring half of the patients to TCN-affiliated clinics and the others to clinics outside of TCN’s network? This should be added for clarity.

Response: Added that the main factor is geographical proximity to these TCN clinics. Patients may not benefit from the services of a TCN-affiliated CHW if there is not a TCN clinic located near them in the community.

  1. You mention in line 323 that April and May 2020 stand out due to being the initial implementation months, but I would also add here that the pandemic was in its infancy in America, with a lot of confusion during that time plus people were probably not being released for medical reasons until at least a little while after that.

Response: Acknowledged with a reference here to the emerging Covid-19 pandemic.

Discussion:

  1. I believe the discussion is missing key information on how something like the Hub can be used in other states in similar situations. More specificity on this point is warranted.

Response:

  1. What has changed with how the Hub deals with patients now that the pandemic is somewhat behind us? A bit more on “where we go from here” both in other states as well as post-pandemic.

Response:

  1. Additionally, will the authors consider a more directed, qualitative analysis with the CHWs and/or other partners and/or patients who were critical in getting this off the ground? That seems to be the next, and clear, area of research to better establish how well the program did or did not work. This would be especially helpful for other DOCs wanting to partner with clinics in their states.

Response:

Minor Edits:

  1. Line 262- I think the word “of” is missing between “summary” and “patient”.

Response: Corrected as suggested

  1. Line 276 & 308- I don’t think you can/should start the sentence with a number.

Response: Corrected as suggested

  1. Appears as if reference 20 and 23 are the same. Same with 11 and 14 and 19. I believe you are only supposed to use the number once when referring to the same reference.

Response: We corrected per ACS style so that each reference is only numbered once (the same number repeated if cited again). These references now appear only once in the reference list.

  1. The thick lines between features of the TCN model (Table 3) are distracting, making the table hard to read. I suggest using lightly dotted lines or listing them as a bulleted list for clarity.

Response: We reformatted this table (now Table 1) for clarity, removing the lines and making it a simple bullet pointed list instead.

  1. The term PCP, listed in Table 4, is never spelled out/explained. Also, what is “X-waivered”? This is also never written out or explained. Both are in also in Table 4. More information on this table in general is warranted.

Response: The abbreviation “PCP” is now defined in the text, prior to referencing in the table (Line __). We also removed reference to the term “x-waivered” which has to do with a provider’s certification to prescribe Suboxone, which is actually no longer a requirement. Instead, we reference “providers who prescribe MOUD”.

Round 2

Reviewer 1 Report

I appreciate the authors' responses and addressing the concerns raised previously. Overall, I think my comments have been addressed adequately. 

Author Response

Thank you for your review of the manuscript entitled The Reentry Health Care Hub: Creating a California-based referral system to link chronically ill people leaving prison to primary care. We appreciate your time and effort during this process. Thank you for your feedback on this paper.

Sincerely,

Bethany Divakaran

Nurse Program Manager

Transitions Clinic Network

[email protected]

and

Natania Bloch

Referrals Coordinator

Transitions Clinic Network

[email protected]

Reviewer 2 Report

Thank you to the authors to responding to all my inquiries and suggestions and making changes to the manuscript, based on that. However, I don't see any responses to the entire "Discussion" section that I had written 3 comments about. Once that is handled, I will feel comfortable accepting the manuscript. 

Author Response

Thank you for your review of the manuscript entitled The Reentry Health Care Hub: Creating a California-based referral system to link chronically ill people leaving prison to primary care. We appreciate your continued review and have addressed the comments from the discussion section that were previously omitted.  

Discussion:

  1. I believe the discussion is missing key information on how something like the Hub can be used in other states in similar situations. More specificity on this point is warranted.

  1. What has changed with how the Hub deals with patients now that the pandemic is somewhat behind us? A bit more on “where we go from here” both in other states as well as post-pandemic.

Response: Added: “Another key next step is to conduct more research on outcomes of these referrals to assess the impact of this intervention for care linkages on engagement in care. A rea-sonable next step is to obtain qualitative input from key stakeholders to the program’s development (nurses, CHWs, leadership and patients) on the impact of this system being in place. Finally, the pandemic required establishing a referral workflow that is done remotely; this referral process could be improved with interventions that more actively engage the patient and their families in planning and creating pre-release connections to the community. This engagement should include collecting data directly from patients, incorporating patient preferences about follow-up care, and increasing connections between patients and the community health system before they are re-leased. The involvement of CHWs with shared lived experience of incarceration and reentry can improve patient engagement in health care, and more research is needed to quantify the dose and type of pre-release interactions that best facilitate post release engagement and improved health outcomes.” This addresses potential next steps now that the Hub can develop further now as the pandemic abates somewhat.

  1. Additionally, will the authors consider a more directed, qualitative analysis with the CHWs and/or other partners and/or patients who were critical in getting this off the ground? That seems to be the next, and clear, area of research to better establish how well the program did or did not work. This would be especially helpful for other DOCs wanting to partner with clinics in their states.

Response: Agreed that further research with CHWs/partners is a reasonable next step to learn more about the impact of this project. I feel like we did address some questions about “what’s next” and replication of this in other contexts at the end of the discussion section (see above comment).

Thank you for your feedback on this paper and we sincerely appreciate your time and energy. We look forward to continuing forward with the publication process.